# Automated Image Analysis of Transmission Electron Micrographs: Nanoscale Evaluation of Radiation-Induced DNA Damage in the Context of Chromatin

**DOI:** 10.3390/cells12202427

**Published:** 2023-10-10

**Authors:** Mutaz A. Abd Al-razaq, Anna Isermann, Markus Hecht, Claudia E. Rübe

**Affiliations:** Department of Radiation Oncology, Saarland University Medical Center, Kirrbergerstr, Building 6.5, 66421 Homburg, Saar, Germany; mutaz.abd-al-razaq@uks.eu (M.A.A.A.-r.);

**Keywords:** automated image analysis, transmission electron microscopy (TEM), heavy ion irradiation, linear energy transfer (LET), DNA damage, DNA double-strand breaks (DSBs), chromatin remodelling

## Abstract

Background: Heavy ion irradiation (IR) with high-linear energy transfer (LET) is characterized by a unique depth dose distribution and increased biological effectiveness. Following high-LET IR, localized energy deposition along the particle trajectories induces clustered DNA lesions, leading to low electron density domains (LEDDs). To investigate the spatiotemporal dynamics of DNA repair and chromatin remodeling, we established the automated image analysis of transmission electron micrographs. Methods: Human fibroblasts were irradiated with high-LET carbon ions or low-LET photons. At 0.1 h, 0.5 h, 5 h, and 24 h post-IR, nanoparticle-labeled repair factors (53BP1, pKu70, pKu80, DNA-PKcs) were visualized using transmission electron microscopy in interphase nuclei to monitor the formation and repair of DNA damage in the chromatin ultrastructure. Using AI-based software tools, advanced image analysis techniques were established to assess the DNA damage pattern following low-LET versus high-LET IR. Results: Low-LET IR induced single DNA lesions throughout the nucleus, and most DNA double-strand breaks (DSBs) were efficiently rejoined with no visible chromatin decondensation. High-LET IR induced clustered DNA damage concentrated along the particle trajectories, resulting in circumscribed LEDDs. Automated image analysis was used to determine the exact number of differently sized nanoparticles, their distance from one another, and their precise location within the micrographs (based on size, shape, and density). Chromatin densities were determined from grayscale features, and nanoparticles were automatically assigned to euchromatin or heterochromatin. High-LET IR-induced LEDDs were delineated using automated segmentation, and the spatial distribution of nanoparticles in relation to segmented LEDDs was determined. Conclusions: The results of our image analysis suggest that high-LET IR induces chromatin relaxation along particle trajectories, enabling the critical repair of successive DNA damage. Following exposure to different radiation qualities, automated image analysis of nanoparticle-labeled DNA repair proteins in the chromatin ultrastructure enables precise characterization of specific DNA damage patterns.

## 1. Introduction

Heavy ion irradiation (IR) has a favorable dose distribution with higher linear energy transfer (LET) and relative biological effectiveness (RBE) compared with photon-based radiotherapy [1]. The enhanced RBE of high-LET versus low-LET IR is driven by unique DNA damage patterns characterized by clustered lesions along particle trajectories that overwhelm the DNA repair capacity of normal and malignant cells [2]. Due to these physical and radiobiological properties, heavy ion IR has strong tumor-killing effects and, at the same time, has the potential to maximally spare normal tissues [3]. Double-strand breaks (DSBs), the most deleterious type of radiation-induced DNA damage, are primarily repaired by two pathways: homologous recombination and non-homologous end-joining (NHE). The choice of which is largely dependent on the cell cycle phase and local chromatin landscape [4]. In non-proliferating cells, circular Ku70/Ku80 heterodimer binds to both ends of individual DSBs and initiates NHEJ at damaged DNA sites, followed by loading of DNA-dependent protein kinase catalytic subunits (DNA-PKcs) onto the DNA-Ku complexes [5]. This DNA-PK holoenzyme then phosphorylates various components of the NHEJ machinery to facilitate final processing and rejoining [5].

The repair of radiation-induced DSBs occurs within a complex chromatin environment. In the undamaged state, chromatin exists in different topological and functional domains that can change dynamically. Based on the degree of compaction (originally defined by contrast staining with basic dyes), chromatin is classified as open, transcriptionally active, gene-rich euchromatin or as more condensed, transcriptionally inert heterochromatin. In response to radiation-induced DSBs, heterochromatic domains are converted to euchromatin, allowing the repair machinery access to areas of damaged DNA. After the DNA damage has been processed and repaired, the original chromatin organization must be restored to ensure cellular functionality. Thus, chromatin responds dynamically to radiation-induced DNA damage by decompacting and expanding, which in turn changes the mobility and accessibility of the damaged locus [6].

Currently, various experimental techniques exist to detect radiation-induced DNA damage, but most methods are not suitable for high-resolution imaging of complex DNA damage patterns in the context of chromatin. Presently, DNA repair markers such as phosphorylated H2AX (γH2AX) and 53-binding protein 1 (53BP1) are visualized as DNA damage foci in DAPI-stained cell nuclei using immunofluorescence microscopy (IFM) [7]. However, because of the limited resolution of conventional IFM, the detailed structure of these DNA damage foci (especially in the case of clustered DSBs after high-LET IR) cannot be examined in detail or within the context of chromatin. To detect DNA-repair proteins within the chromatin ultrastructure, we established immunogold-labeling techniques using transmission electron microscopy (TEM) [8,9,10]. The nanometer resolution of TEM permits the visualization of repair proteins at the single-molecule level in different chromatin compartments [11]. In previous IFM and TEM analyses, we characterized DNA damage patterns in human fibroblasts following low-LET and high-LET IR, respectively [8,12]. After high-LET IR, densely clustered DNA damage in areas of low electron density domains (LEDDs) was detectable using TEM [13]. However, counting individual nanoparticles and determining clustered lesions using electron microscopy was extremely time-consuming, and unambiguous assignment to euchromatic or heterochromatic compartments with exact demarcation of LEDDs was not always possible. In addition, as quantitative evaluations become the standard in biological research, the need for higher data acquisition throughput in TEM imaging also grows. Therefore, we have established the automated analysis of TEM images using existing AI-based software tools to effectively enable the automatic detection of individual nanoparticles based on their specific characteristics (size, round shape, electron density) and their precise localization in different chromatin densities. Here, we used these advanced image analysis techniques to assess the DNA damage pattern in human fibroblasts following high-LET IR.

## 2. Materials and Methods

Cell Culture: Human dermal fibroblasts (PromoCell, Heidelberg, Germany) were grown on coverslips in Fibroblast Growth Medium (PromoCell) at 37 °C and 5% CO_2_. Confluent interphase cells (with homogenous chromatin density) were used for experiments.

Low-LET and high-LET irradiation: Low-LET IR with 6-MV photons (10 Gy; dose-rate 2 Gy/min) was performed at the Department of Radiation Oncology, Saarland University (Homburg, Germany) using the linear accelerator Artiste™ (Siemens, München, Germany). High-LET IR with carbon ions (9.5 MeV/nucleon; LET 190 keV/µm; fluence 5× 10^6^ particles/cm^2^; calculated mean dose: 1.52 Gy) was performed at GSI Center for Heavy Ion Research (Darmstadt, Germany) using the UNILAC accelerator as previously described [13]. Cells were analyzed at different time points after low-LET or high-LET IR and compared with sham-irradiated cells.

Immunofluorescence microscopy analysis: Cells were fixed in 4% paraformaldehyde and permeabilized in 0.2% Triton X-100. After blocking with 1% BSA in PBS overnight, cells were incubated with primary antibodies (anti-53BP1, Novus Biologicals, Wiesbaden Nordenstadt, Germany; anti-pKu70, Abcam, Cambridge, UK, anti-pKu80, LifeSpan Biosciences, Seattle, WA, USA, anti-DNA-PKcs, Novus Biologicals, Wiesbaden Nordenstadt, Germany), followed by AlexaFluor-488 or AlexaFluor-568 secondary antibodies (Invitrogen, Karlsruhe, Germany). Finally, cells were mounted in VECTAshield^TM^ mounting medium with 4′,6-diamidino-2-phenylindole (DAPI, Vector Laboratories, Burlingame, CA, USA). Fluorescence images were captured using a Nikon-Eclipse Ni fluorescence microscope equipped with a charge-coupled device camera and acquisition software (Nikon, Düsseldorf, Germany). Foci numbers were quantified, and foci areas were measured at an objective magnification of 60× until at least 50 cells were registered per sample.

Transmission Electron Microscopy analysis: Cells were fixed with 2% paraformaldehyde and 0.05% glutaraldehyde in PBS. Fixed samples were dehydrated using increasing concentrations of ethanol and infiltrated with LR White resin overnight (Plano, Wetzlar, Germany). Subsequently, samples were embedded in fresh resin with an accelerator at 37 °C until the resin polymerized. Ultrathin sections (70 nm) were cut on a Microtome Ultracut UCT (Leica, Wetzlar, Germany) with diamond knives (Diatome, Biel, Switzerland), gathered on pioloform-coated nickel grids, and processed for nanoparticle-labelling. To block nonspecific staining, sections were placed on drops of blocking solution (Aurion, Wageningen, The Netherlands). Afterward, sections were rinsed and incubated with primary antibodies (anti-53BP1, Novus Biologicals, Wiesbaden Nordenstadt, Germany; anti-pKu70, Abcam, Cambridge, UK, anti-pKu80, LifeSpan Biosciences, Seattle, WA, USA, anti-DNA-PKcs, Novus Biologicals, Wiesbaden Nordenstadt, Germany), overnight at 4 °C. After washing, secondary antibodies conjugated with 6 nm or 10 nm gold particles (Aurion, Wageningen, The Netherlands) were applied to the sections for 1.5 h. Sections were then rinsed and fixed with 2% glutaraldehyde in PBS. All sections were stained with uranyl acetate and examined using Tecnai Biotwin™ transmission electron microscope (FEI, Eindhoven, The Netherlands). For each radiation quality (low-LET photons, high-LET carbons), single- and clustered nanoparticles were counted in ≥25 randomly chosen nuclear sections per examination time. The number and area of LEDDs were also measured in ≥25 cell nuclei.

Automated image analysis of transmission electron micrographs: Nuclear sections were systematically scanned at sufficient resolution to identify cells with LEDDs. Manual annotations were made for each micrograph to avoid artifacts and non-relevant areas. Acquired micrographs with LEDDs were edited by adjusting the contrast to enhance nanoparticle detection. For automatic annotation of gold nanoparticles, the brightness (50% increase) and contrast (80% increase) adjustments were uniformly applied to all original TEM images. Regions of interest were segmented and analyzed using a well-trained AI classifier using HALO^®^ Image Analysis Platform version 3.4.2986 (Indica Labs, Albuquerque, NM, USA). The software was able to automatically identify and select LEDD areas using the Area Quantification module v2.4.2, and LEDD areas were further subdivided into different regions related to the LEDD border. All nanoparticles around the LEDDs were detected and counted automatically, followed by the calculation of the nanoparticle distribution and clusters per unit area using the spatial plot tool.

Statistical analysis: GraphPad Prism (version 9.4.1, GraphPad Software, San Diego, CA, USA) was used to analyze data. Data were presented as the mean of at least three experiments ± SE. Two-way ANOVA (multiple comparisons) and the Mann–Whitney Test were used for estimating the differences among groups, followed by multiple comparisons between data sets. A *p*-value of <0.05 was considered statistically significant, <0.01 as highly statistically significant, and <0.001 as exceptionally statistically significant. In the figures, statistically significant differences are indicated as asterisks directly above the bar when comparing to the previous time point and as asterisks above brackets when comparing between two different study groups (* *p* < 0.05, ** *p* < 0.01, and *** *p* < 0.001).

## 3. Results

### 3.1. Clustered Foci and Decondensed Chromatin Regions following High-LET IR

In earlier work, fluorescence microscopy was used to show that exposure of cell nuclei to heavy carbon ions leads to spatially defined DNA damage along the particle trajectories [14,15]. To study the initial formation and subsequent repair of this clustered DNA damage, cell monolayers were irradiated with vertical beam direction (90° angle to the monolayer plane). Using IFM, DNA damage-induced foci appeared within the nuclei 0.5 h after high-LET IR, generating small co-localizing foci for pKu70 and pKu80 and larger clustered foci for 53BP1 and pKu80 (Figure 1A, left panel). Radiation-induced foci formed after high-LET IR are brighter and larger than those generated following low-LET IR, likely because many DSBs are induced within these particle trajectories. Using IFM, the number and area of 53BP1 and pKu80 foci per cell nucleus were quantified at different time points after high-LET IR (Figure 1A, right panel). The mean number of clustered foci per cell increased until 0.5 h post-IR (53BP1: 5.00 ± 0.13 foci/cell; pKu80: 4.62 ± 0.11 foci/cell) and subsequently decreased within 24 h post-IR (53BP1: 2.00 ± 0.26 foci/cell; Ku80: 2.34 ± 0.18). The maximal track number 0.5 h after irradiation with vertical beam direction correlates with the applied particle fluence of 5 × 10^6^ particles/cm^2^. Moreover, measuring the mean area of clustered foci after vertical beam direction, we observed an increase in the mean track area over time, with the maximum at 5 h post-IR (53BP1: 2.28 ± 0.14 µm^2^; pKu80 1.33 ± 0.09 µm^2^) and with slight reductions at 24 h post-IR.

Subsequently, TEM imaging was used to characterize the ultrastructural pattern of DNA damage caused in cells exposed to high-LET IR. For TEM analysis, irradiated cells were harvested as monolayers to preserve the structural organization of the nucleus, as well as the beam direction in the embedded cells. Cell samples were stained with uranyl acetate to enhance the contrast of chromatin, and defined levels of gray were assigned to euchromatin and heterochromatin. Detecting multiple repair factors within the chromatin ultrastructure using TEM requires the selective use of gold-conjugated antibodies with varying particle sizes (10-nm or 6-nm nanoparticles), which were subsequently colored in the micrograph for better visualization. Non-irradiated fibroblasts are characterized by a homogenous chromatin organization throughout the entire nucleus. Following low-LET IR, no distinct local or global changes in the nanostructural chromatin organization were detectable during the DNA repair process. Following high-LET IR with vertical beam direction, varying numbers of electron-lucent regions were observed, likely reflecting LEDDs following particle transversals. In an earlier study, we were able to show that these LEDDs in TEM reflect the clustered foci in IFM [13]. Subsequently, DNA repair proteins (pKu70, pKu80, and 53BP1) were labeled with immunogold beads, and these nanoparticles were visualized in the chromatin ultrastructure using TEM. Our results show that pKu70 (blue) and pKu80 (red) were predominantly located at the border of LEDDs, while 53BP1 (green) was also found in heterochromatic domains within LEDDs. A phosphorylated Ku70/80 heterodimer is required for efficient repair of radiation-induced DSBs via the NHEJ pathway, with a single Ku70/80 heterodimer binding each of the two broken ends of the DSB. Our findings show that compared with low-LET IR, where only isolated DNA damage was detectable in the entire nucleus, there is a high concentration of actively processed DSBs around the LEDDs of high-LET irradiated cells.

### 3.2. Quantification of pKu80 and DNA-PKcs following Low-LET versus High-LET IR

In the early stage of the NHEJ process, DNA-PKcs is recruited to Ku70/Ku80 heterodimer-bound DNA ends. To monitor DSB repair within the chromatin ultrastructure of human fibroblasts following low-LET versus high-LET IR, nuclear sections were immunogold-labeled for pKu80 and pDNA-PKcs (6-nm versus 10-nm particle size), and these nanoparticles were counted in defined euchromatic and heterochromatic regions (Figure 2A).

Following low-LET IR, the mean number of radiation-induced pKu80 and DNA-PKcs nanoparticles decreased from 0.5 h post-IR to 5 h post-IR (pKu80: 38.7 ± 5.1 to 26.6 ± 2.3 nanoparticles; DNA-PKcs: 19.8 ± 3.3 to 7.7 ± 1.8 nanoparticles), indicating slower, but efficient, repair kinetics in heterochromatic compartments. By contrast, following high-LET IR, the mean number of pKu80 and DNA-PKcs nanoparticles increased from 0.5 h post-IR to 5 h post-IR (pKu80: 25.2 ±3.4 to 69.5 ± 7.2 nanoparticles; DNA-PKcs: 28.3 ± 2.8 to 47.2 ± 9.6 nanoparticles) (Figure 2A). While euchromatic regions revealed a slight decrease in pKu80 and DNA-PKcs nanoparticles with time, the heterochromatin, however, showed a significant increase. Based on these data, we hypothesize that the delay in the detection of DSBs clustered in heterochromatic compartments induced by high-LET IR is due to essential chromatin remodeling.

To further characterize the DNA-damage patterns caused by low-LET versus high-LET IR, we quantified the clusters for pKu80 and DNA-PKcs (subdivided into size categories 1–2, 3–4, >4 beads per cluster) separately in euchromatic and heterochromatic regions (Figure 2B). Following low-LET and high-LET IR, most of the pKu80 clusters consisted of two beads separated by an almost constant distance, reflecting each pKu80 molecule bound to the free ends of single breaks (>90–100%). In heterochromatin, the number of complex DSBs (≤3 beads) already increased at 0.5 h (~5%) after high-LET IR and further increased at 5 h (10%). This trend was even more impressive for DNA-PKcs. 5 h after high-LET IR, almost 30% of heterochromatic lesions were clustered DSBs (≤3 nanoparticles) (Figure 2B). Together, these findings indicate that high-LET IR induces clustered DNA lesions in heterochromatic regions, and this damage level even increased over time, suggesting that DSB clustering in heterochromatin following high-LET IR perturbs efficient repair.

### 3.3. Detection of Nanoparticles in the Areas of LEDDs Using Automated Image Analysis

After loading the TEM images into the HALO^®^ Image Analysis platform and adjusting the scale for the micrographs, LEDDs were automatically segmented based on electron density, and the area of the LEDDs was calculated (Figure 3A). With this automated segmentation, 63 LEDDs were delineated following high-LET IR, with an averaged area of 0.76 ± 0.22 at 0.5 h post-IR and 1.65 ± 0.44 at 5 h post-IR (Figure 3A, right lower panel), correlating with the areas measured for clustered foci using IFM (Figure 1A). To capture the distribution of DNA repair factors associated with chromatin remodeling, the areas inside, outside, and at the border of the LEDDs were defined (each at 300 nm intervals) and segmented accordingly (Figure 3A, left lower panel). After appropriate contrast adjustment, nanoparticles were automatically recognized and quantified for the various repair factors (pKu80 and DNA-PKcs). All nanoparticles were automatically assigned to the corresponding region so that the spatial distribution in relation to the LEDD could be determined for each repair factor (Figure 3A, right panel). Figure 3B shows the quantitative evaluation of heterochromatic pKu80 and DNA-PKcs clusters (≥3 nanoparticles) in relation to the LEDD region after 0.5 h and 5 h post-IR. A shift in cluster size as a function of time was observed for both repair factors. At the early time point (0.5 h post-IR), only a few clusters were detected inside the LEDDs (pKu80: 2.8% ± 0.78%; pDNA-PKcs: 4.9% ± 1.18%), a little more in the border region (pKu80: 15.8% ± 1.49%; DNA-PKcs: 16.0% ± 2.77%) and most outside the LEDDs (pKu80: 56.6% ± 2.75%; 53.9% ± 3.58%). By contrast, 5 h post-IR, most of the pKu80 and DNA-PKcs clusters were observed in the border areas of LEDDs (pKu80: 45.6% ± 1.44%; DNA-PKcs: 55.8% ± 2.55%) (Figure 3B). Along with the observed enlargement of LEDDs, our findings suggest that the chromatin within the particle trajectory area progressively opened to allow the repair of clustered DNA damage.

### 3.4. Precise Analysis of the LEDD Boundaries Using Automated Image Analysis

To investigate the opening of chromatin in relation to the density of 53BP1 nanoparticles in more detail, we focused on the border areas of LEDDs. Based on the original TEM micrograph, the chromatin density was defined according to grayscale features and categorized into euchromatin versus heterochromatin (Figure 4A). Subsequently, high-LET-IR-induced LEDDs were delineated using automated segmentation. The nanoparticles within the LEDDs and the border regions, as well as beyond, were also segmented (Figure 4B, left panel). The spatial distribution of nanoparticles, especially regarding cluster formation, was visualized with the density heat map module. This density map clearly shows that most of the 53BP1 nanoparticles were located in the immediate border area of the LEDDs, and the formation of clustered lesions mainly occurs in heterochromatic areas (Figure 4B, right panel: ≥4 beads →orange-red areas). Dividing the boundary regions around segmented LEDDs into four different distance ranges (1–100 nm, 101–200 nm, 201–300 nm, 301–400 nm), our results showed that the density of 53BP1 nanoparticles decreases significantly with increasing distance from the LEDD (Figure 4C). These results suggest that radiation-induced DSBs lead to an opening of the chromatin and, thus, to an increasing enlargement of the LEDDs following high-LET IR.

## 4. Discussion

In living organisms, the biological outcome of IR exposure is determined by the spatial and temporal distribution of ionization and excitation events, leading to the occurrence of different types of complex DNA damage. Approaches such as nanodosimetry/microdosimetry and Monte Carlo track-structure simulations have been successfully adopted to describe radiation quality effects. However, physical features alone are not sufficient to assess the extent and complexity of radiation-induced DNA damage. The latter is the result of an interplay between radiation trace structure and spatial chromatin and depends on the dynamic response of chromatin, affecting the activation and efficiency of the DNA repair machinery [16]. In future collaborative projects, our experimental TEM results will be compared with the biophysical simulation code PARTRAC for stochastic modeling of DSB repair after photon and ion IR. PARTRAC combines track structure calculations with DNA models on diverse genomic scales and, therefore, enables the prediction of DNA damage yields and patterns for various radiation qualities.

The aim of the present study was to establish the automated image analysis of transmission electron micrographs to investigate patterns of radiation-induced DNA damage within the cell nucleus on the nanometer scale. Here, we show that automated image analysis of nanoparticles in TEM micrographs enables precise characterization of complex DNA damage in combination with chromatin architecture and dynamics [17]. With powerful analysis functions and fast processing speeds, this automation strategy enables the in-depth characterization of radiation damage with high sample throughput so that the specific DNA damage pattern following exposure to different radiation qualities can be recorded not only qualitatively but also quantitatively [18]. Based on their circumscribed size, shape, and density, the exact number of the different-sized nanoparticles, their distance to each other, as well as their exact localization in the micrographs can be determined with appropriate image analysis software. With the tissue classifier module, we used a state-of-the-art machine learning algorithm to identify chromatin densities based on grayscale features and categorized them into euchromatic or heterochromatic regions. Automating nuclear segmentation to delineate LEDDs across entire sections eliminated the need to manually draw outlines of areas of interest, increasing the objectivity of the analysis. In addition, we used different spatial analysis tools to identify the proximity and relative spatial distribution of nanoparticles throughout the nuclear domains. Using the spatial density heatmap analysis algorithm, we were able to measure and calculate the average density of nanoparticles within a certain distance of segmented LEDDs, and corresponding proximity histograms were automatically generated. Overall, our earlier TEM results were confirmed using this automation strategy. Moreover, this AI-based image analysis not only enables significantly faster data generation but also a far more precise analysis of the radiation-induced DNA damage pattern [19]. Visual TEM evaluation already highlighted clear differences in the DNA damage pattern and DNA repair capacity between low-LET and high-LET IR [20]. The automated image analysis of human fibroblasts at different times after low-LET versus high-LET IR enables the efficient evaluation of nanoparticle-labeled DNA repair proteins in the context of chromatin ultrastructure.

Radiation-induced DSBs have major effects on cell biology of transcription, replication, and interface with metabolic responses. Accurate recognition and timely repair of DSBs in complex chromatin environments requires a tightly coordinated DNA damage response (DDR). To detect DSBs in cell nuclei, IFM is generally used for visualization of γH2AX or other radiation-induced foci. However, the resolution of standard fluorescence microscopy is too low to detect individual proteins at the single-molecule level, so DNA repair events cannot be linked to other DDR mechanisms. Our previous TEM studies with nanoscale-resolution imaging of accumulated DNA damage after high-LET IR revealed intriguing new insights into DSB processing within the chromatin environment. The basic idea of this TEM study was not only to go beyond the resolution of IFM but to systematically record and evaluate the various repair factors related to the chromatin status with a feasible time commitment. Automated image analysis of TEM micrographs offers an unbiased approach to investigating DDR by measuring protein localizations, interactions, and concentrations in the ultrastructure of the cell nucleus.

In this study, we used the computational pathology software HALO AI (version 3.4.2986; Indica Labs, Albuquerque, NM, USA) to evaluate TEM micrographs. HALO AI image analysis platform is specialized for quantitative analysis in digital pathology, enabling segmentation using artificial intelligence. User-friendly, intuitive HALO modules for different applications improve image processing speed and permit transparent workflows. Here, we used automatic segmentation of differently sized nanoparticles in combination with spatial analysis to analyze precisely immunogold-labeling patterns in nuclear sections, thereby characterizing the DNA damage pattern after exposure to different radiation qualities.

In traditional post-embedding TEM experiments, cells are chemically fixed, dehydrated, and embedded in resins. Resin blocks containing the specimen are then sectioned into thin slices to ensure the collection of the electron beam after passing through the sample. Since biological specimens (cells and tissue) are composed of elements with low atomic numbers, the difference in electron density is small, resulting in low-contrast images. To increase specimen contrast, biological samples are traditionally stained with heavy metal salts, such as osmium tetroxide, lead citrate, and uranyl acetate. Osmium interacts with lipids, uranium binds to phosphate and amino groups, and lead interacts with negatively charged groups. Overall, these metallic dyes enable differential staining of organelles and compartments in mammalian cells. To ensure meaningful comparisons of the complex organization within cells at the ultrastructural level, all samples must be processed in exactly the same way. Moreover, threshold values for determining LEDDs must not be changed in original TEM images; otherwise, the shape and area of LEDDs would have been variable depending on the setting. Another important point is that uranyl acetate contrasting is not specific to DNA and, therefore, is not a reliable marker for DNA compactness in TEM imaging applications. In future projects, systematic investigations are planned to analyze the pathophysiological significance of these LEDDs using immunogold labeling for euchromatic and heterochromatic histone modifications.

Post-embedding immunoelectron microscopy is a powerful method for detecting antigens on the surface of sections. Nanoparticle-antibody conjugates with defined structure and stoichiometry are indispensable tools for subcellular mapping in high-resolution TEM. Electron microscopic imaging exploits the high electron density of gold (19.3 g/mL) compared with that of proteins (1.35 g/mL), providing electron opacity and high contrast to biological materials, and thus guarantees reliable detection during visual or automated evaluation. Due to its particulate and countable nature, colloidal gold is preferred as an antibody label as it offers the ability to quantify the concentration of antigens. However, the specificity and affinity of the antibody can influence nanoparticle quantification measurements. Accordingly, the number of nanoparticles cannot be directly derived from the number of antigens on the sample section because the labeling efficiency is influenced by various physical, chemical, and biological factors, mainly arising from sample preparation. However, constant and reproducible labeling efficiencies achieved by standardizing processing conditions are usually considered sufficient for quantification purposes. In our study, gold nanoparticles carried a single binding site for the primary antibody. Therefore, we hypothesize that although relative immunogold labeling does not provide the exact antigen concentration, it allows direct comparisons between subcellular site concentrations.

Our results provide direct evidence that high-LET IR induces clustered DNA damage along particle trajectories through highly focused ionization events, leading to extensive chromatin remodeling with the formation of LEDDs. In these particle trajectories, DSBs are increasingly detectable during this chromatin remodeling process, preferentially in the border areas between euchromatin and heterochromatin. Our results suggest that high-LET IR, in contrast to low-LET IR, is associated with pronounced chromatin relaxation in the form of LEDDs to enable the critical repair process of clustered DNA damage. The increase in heterochromatic DSBs and the sustained chromatin decondensation in the form of LEDDs indicates that cells are unable to repair the accumulated DNA damage and restore the original chromatin organization after high-LET IR.

## 5. Conclusions

Highly clustered DNA lesions, generated by extremely localized energy deposition of high-LET IR, pose a serious threat to cell viability by compromising both genomic and epigenomic integrity. A better understanding of this coordination between repairing DNA damage and restoring original chromatin structures will advance our view of genomic and epigenomic maintenance in response to DNA damage [13]. Overall, this automation strategy for quantifying nanoparticles in the chromatin context significantly reduces workload and enables comparative studies to evaluate dose distributions on micro- and nanometer scales following exposure to different radiation qualities. The increased throughput provided using automated acquisition schemes and the resulting generation of large amounts of data opens new possibilities for quantitative TEM studies in radiation research.

## Figures and Tables

**Figure 1 cells-12-02427-f001:**
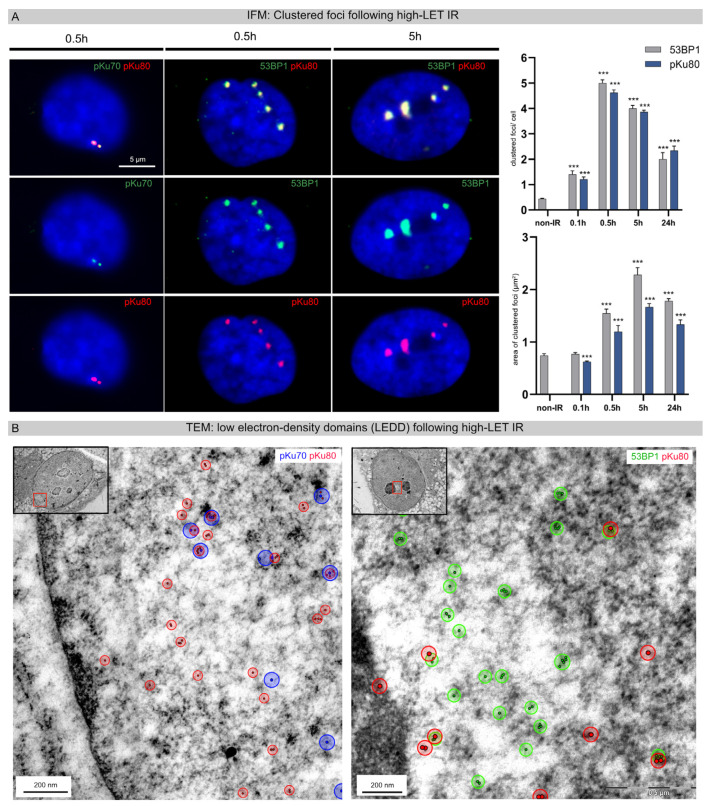
Clustered foci and decondensed chromatin regions (LEDDs) following high-LET IR. (**A**) IFM micrographs of DAPI-stained nuclei double-stained for pKu70 (green) with pKu80 (red) or 53BP1 (green) with pKu80 (red), analyzed at 0.5 h and 5 h after high-LET IR (vertical beam direction). For 53BP1 and pKu80, the number and area of clustered foci were quantified at 0.1 h, 0.5 h, 5 h, and 24 h after high-LET IR. (**B**) TEM micrographs of nanoparticle-labeled pKu70 (blue), pKu80 (red), and 53BP1 (green) in the chromatin ultrastructure. Within 5 h after high-LET IR, multiple pKu70 (blue), pKu80 (red), and 53BP1 nanoparticles occasionally formed clusters distributed inside and outside LEDDs. Insets: overview images of the nucleus; framed regions are shown at higher magnification. Statistically significant differences are indicated as asterisks directly above the bar when comparing to the previous point of investigation (*** *p* < 0.001).

**Figure 2 cells-12-02427-f002:**
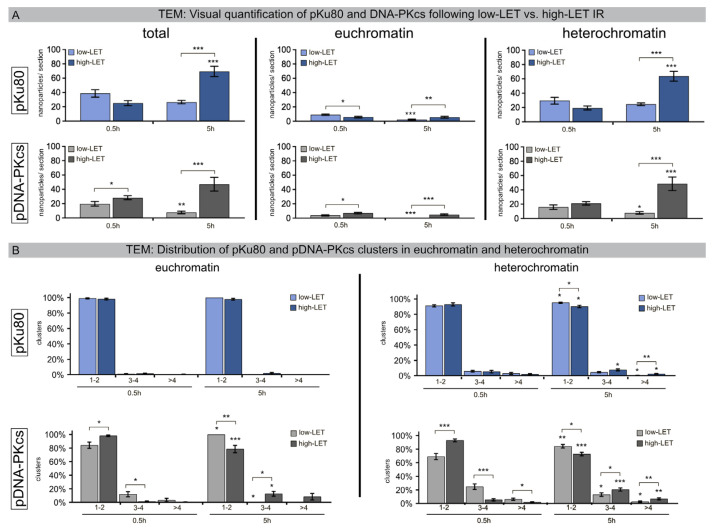
Quantification of pKu80 and DNA-PKcs following low-LET versus high-LET IR. (**A**) The mean number of nanoparticles labeling pKu80 and DNA-PKcs (per nuclear section) in euchromatin and heterochromatin. (**B**) Percentage of clusters (subdivided into 1–2, 3–4, and >4 categories) in euchromatin and heterochromatin quantified in nuclear sections using TEM at 0.5 h and 5 h after low-LET versus high-LET IR. Statistically significant differences are indicated as asterisks directly above the bar when comparing to the previous time point and as asterisks above brackets when comparing between two different study groups (* *p* < 0.05, ** *p* < 0.01, and *** *p* < 0.001).

**Figure 3 cells-12-02427-f003:**
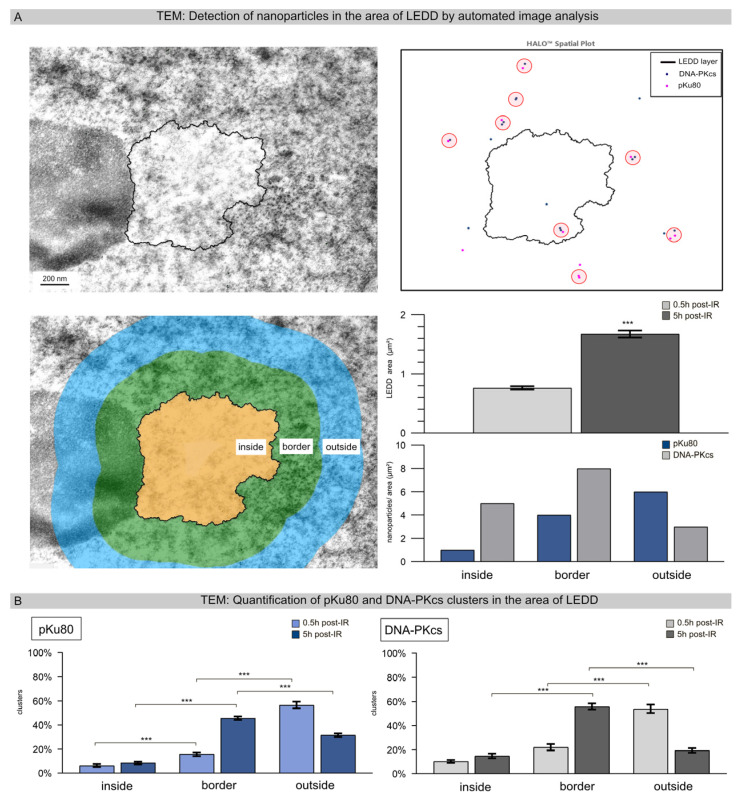
Detection of nanoparticles in the areas of LEDDs using automated image analysis. (**A**) TEM micrographs 5 h after high-LET IR show an LEDD next to the nucleolus. Using the spatial analysis tool of the HALO software, version 3.4.2986 (Indica Labs, Albuquerque, NM, USA), the LEDD was automatically segmented, and the differently-sized nanoparticles were detected. For further characterization of the repair factors in relation to the LEDD, the areas inside and outside the LEDD, as well as the border area, were defined, and the number of pKu80 and DNA-PKcs nanoparticles was automatically quantified. The area of each LEDD was quantified at 0.5 h and 5 h after high-LET IR. (**B**) Quantification of pKu80 and DNA-PKcs clusters inside and outside the LEDD, as well as in the border region, analyzed at 0.5 h and 5 h after high-LET IR. Statistically significant differences are indicated as asterisks directly above the bar when comparing to the previous time point and as asterisks above brackets when comparing between two different study groups (*** *p* < 0.001).

**Figure 4 cells-12-02427-f004:**
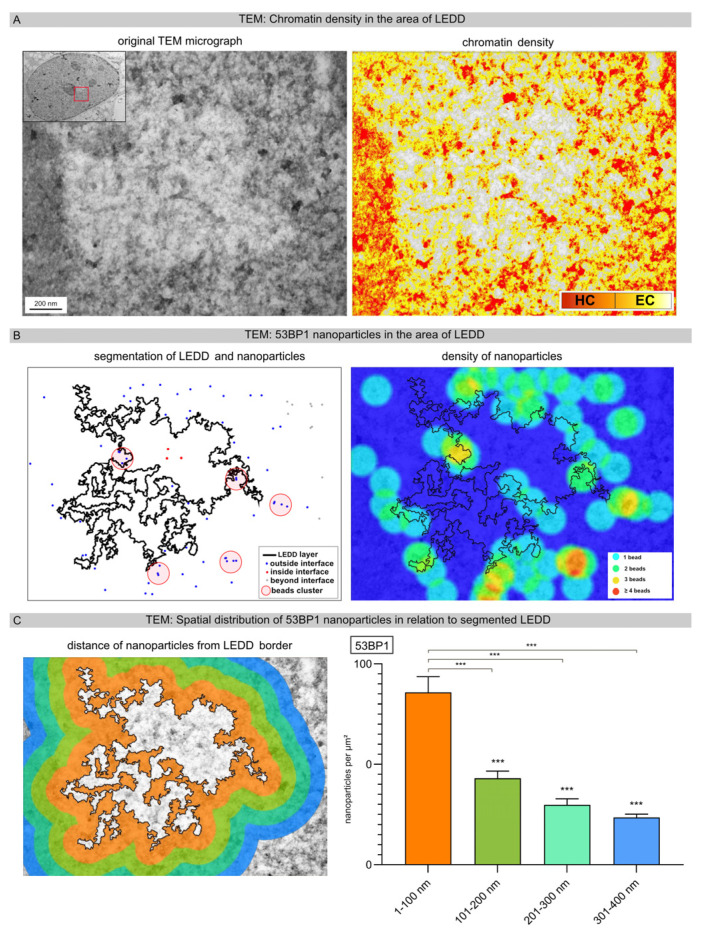
Precise analysis of LEDD boundaries using automated image analysis. (**A**) Based on the original TEM micrograph, chromatin density was defined according to grayscale features and categorized into euchromatin versus heterochromatin. (**B**) High-LET-IR-induced LEDDs and nanoparticles were delineated using automated segmentation. The spatial distribution of the nanoparticles, particularly regarding cluster formation, was visualized as a density heatmap. (**C**) Different boundary regions around the high-LET-IR-induced LEDDs were defined, and the spatial distribution of nanoparticles relative to segmented LEDDs was determined. In the inset, the red square shows the enlarged image section of the entire cell nucleus. Statistically significant differences are indicated as asterisks directly above the bar when comparing to the previous point of investigation (*** *p* < 0.001).

## Data Availability

The data presented in this study are available on request from the corresponding author.

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
