# Peer review of "Automated Image Analysis of Transmission Electron Micrographs: Nanoscale Evaluation of Radiation-Induced DNA Damage in the Context of Chromatin"

_cells, 2023, doi:10.3390/cells12202427_

Round 1

Reviewer 1 Report

The manuscript of Abd Al-razaq presents the results of a quantitative analysis of the distribution of factors involved in DNA repair following radiation damage induced by low-LET photons and high-LET carbon ions. To analyze the distribution of these factors, the authors use post-embedding immune-gold labeling. Gold particles were manually annotated, and their distribution was correlated with the local electron density within sections of cell nuclei, particularly focusing on low electron density domains formed after clustered DNA damage repair. For segmenting these low-density regions in images, AI-based classifiers were used. The authors have obtained important findings regarding the compartmentalization of these low-density  domains: while chromatin-binding marker 53BP1 is distributed within regions of low density, pKu complex is associated with the surrounding regions of a higher density.

As I understand, the main objective of the study was to correlate the distribution of gold labeling with the compaction/activity status of chromatin (eu- and heterochromatin). Unfortunately, this objective cannot be achieved with the methodological approach used by the authors. Uranyl acetate contrasting is not specific to DNA and cannot be not a reliable marker for DNA compactness, as it also binds to RNA and protein components in the nucleus. In the context of clustered DNA damage, this was recently demonstrated too (https://pubmed.ncbi.nlm.nih.gov/32168789/). To interpret the chromatin status, the authors should have used DNA-specific contrast methods or immunogold labeling probes specific to histone modifications characteristic for euchromatin or heterochromatin. Therefore, this article, in its current form, cannot be accepted for further consideration.

As a possible solution, I can suggest that the authors reorganize their interpretation without relation on chromatin status (e.g., linking the factor to the overall electron density of nuclear contents) or to introduce DNA or chromatin-specific techniques.

Specific comments:

Comparison with stochastic distribution models has to be always used to address the significance of the gold particle distributions in the real data.

130-131. The exact number of nuclei and their total area have to be mentioned.

 136. The contrast is essential for specificity of AI-based segmentation. How was it changed?

 232-234. Interpretation of gold clustering has to take into account "amplification" artefacts resulting from binding of several (usually two) secondary antibodies to a single primary one.

 284. Arbitrary thresholding is prone to bias. The shape and the area can strongly vary depending on the settings used. This part has to be explained in more detail.

Author Response

Reviewer #1

Comments and Suggestions for Authors

The manuscript of Abd Al-razaq presents the results of a quantitative analysis of the distribution of factors involved in DNA repair following radiation damage induced by low-LET photons and high-LET carbon ions. To analyze the distribution of these factors, the authors use post-embedding immune-gold labeling. Gold particles were manually annotated, and their distribution was correlated with the local electron density within sections of cell nuclei, particularly focusing on low electron density domains formed after clustered DNA damage repair. For segmenting these low-density regions in images, AI-based classifiers were used. The authors have obtained important findings regarding the compartmentalization of these lowdensity domains: while chromatin-binding marker 53BP1 is distributed within regions of low density, pKu complex is associated with the surrounding regions of a higher density. As I understand, the main objective of the study was to correlate the distribution of gold labeling with the compaction/activity status of chromatin (eu- and heterochromatin). Unfortunately, this objective cannot be achieved with the methodological approach used by the authors. Uranyl acetate contrasting is not specific to DNA and cannot be not a reliable marker for DNA compactness, as it also binds to RNA and protein components in the nucleus. In the context of clustered DNA damage, this was recently demonstrated too (https://pubmed.ncbi.nlm.nih.gov/32168789/). To interpret the chromatin status, the authors should have used DNA-specific contrast methods or immunogold labeling probes specific to histone modifications characteristic for euchromatin or heterochromatin. Therefore, this article, in its current form, cannot be accepted for further consideration. As a possible solution, I can suggest that the authors reorganize their interpretation without relation on chromatin status (e.g., linking the factor to the overall electron density of nuclear contents) or to introduce DNA or chromatin-specific techniques.

The main objective of the study was to establish automated image analysis to carry out the extremely time-consuming counting of gold beads of different sizes, as well as their assignment to one another and their relationship to thecompaction status of chromatin in a time-efficient manner and regardless of potential bias of investigators. However, we agree that uranyl acetate contrasting is not specific to DNA and therefore is not a reliable marker for DNA compactness in TEM imaging applications. Therefore, in the revised paper we will not interpret these low electron density regions formed after clustered DNA damage in relation to the chromatin status and will refer to them as so-called low electron density domains (LEDD). In future projects we plan to carry out systematic investigations analyzing the pathophysiological significance of these less contrasted regions using immunogold-labeling for euchromatic and heterochromatic histone modifications.

Line 396-413: “In traditional post-embedding TEM experiments, cells are chemically fixed, dehydrated and embedded in resins. Resin blocks containing the specimen are then sectioned into thin slices to ensure collection of the electron beam after passing through the sample. Since biological specimens (cells and tissue) are composed of elements with low atomic numbers, the difference in electron density is small, resulting in low-contrast images. To increase specimen´s contrast, biological samples are traditionally stained with heavy metal salts, such as osmium tetroxide, lead citrate and uranyl acetate. Osmium interacts with lipids, uranium binds to phosphate and amino groups, and lead interacts with negatively charged groups. Overall, these metallic dyes enable differential staining of organelles and compartments in mammalian cells. To ensure meaningful comparisons of the complex organization within cells at the ultrastructural level, all samples must be processed exactly in the same way. Moreover, threshold values ​​for determining LEDDs must not be changed in original TEM images, otherwise the shape and area of LEDDs would have been variable depending on the setting. Another important point is that uranyl acetate contrasting is not specific to DNA and therefore is not a reliable marker for DNA compactness in TEM imaging applications. In future projects systematic investigations are planned to analyze the pathophysiological significance of these LEDDs using immunogold-labeling for euchromatic and heterochromatic histone modifications.”

Specific comments: Comparison with stochastic distribution models has to be always used to address the significance of the gold particle distributions in the real data. 130-131.

Data on the precise distribution of nanoparticles in cell nuclei following IR exposure are only available in low amounts hitherto but could help to increase our knowledge concerning radiation damage induction. In future collaborative projects, our experimental TEM results will be compared with the biophysical simulation code PARTRAC for stochastic modeling of DSB repair after photon and ion IR. PARTRAC combines track structure calculations with DNA models on diverse genomic scales, and therefore enables the prediction of DNA damage yields and patterns for various radiation qualities. Even if the comparison with experimental results shows that further model refinements are required, the present PARTRAC model already represents a promising step towards system modeling of the cellular response to radiation.

Line 334-439: “In future collaborative projects, our experimental TEM results will be compared with the biophysical simulation code PARTRAC for stochastic modeling of DSB repair after photon and ion IR. PARTRAC combines track structure calculations with DNA models on diverse genomic scales, and therefore enables the prediction of DNA damage yields and patterns for various radiation qualities.” 

The exact number of nuclei and their total area have to be mentioned. 136.

Line 140-142: “For each radiation quality (low-LET photons, high-LET carbons), single- and clustered nanoparticles were counted in ≥25 randomly chosen nuclear sections per examination time. The number- and area of LEDDs were also measured in ≥25 cell nuclei.”

Line 271-273: “With this automated segmentation, 63 LEDDs were delineated following high-LET IR, with an averaged area of 0.76 ±0.22 µm2 at 0.5h post-IR and 1.65 ±0.44 µm2 at 5h post-IR.”

The contrast is essential for specificity of AI-based segmentation. How was it changed? 232-234.

Since metallic gold has a high atomic number (Z=79), gold nanoparticles have a high electron density and therefore provide opacity to the electron beam. The physicochemical properties of these spherical nanoparticles are very different from those of the bulk material in cells and tissues and can therefore be easily determined based on their size and shape in different biomedical applications. For automatic annotation of gold nanoparticles in our study, the brightness (50% increase) and contrast (80% increase) adjustments were uniformly applied to all original TEM images.

Line 147-149: “For automatic annotation of gold nanoparticles, the brightness (50% increase) and contrast (80% increase) adjustments were uniformly applied to all original TEM images.”

Interpretation of gold clustering has to take into account "amplification" artefacts resulting from binding of several (usually two) secondary antibodies to a single primary one. 284.

Line 415-432: “Post-embedding immunoelectron microscopy is a powerful method for detecting antigens on the surface of sections. Nanoparticle-antibody conjugates with defined structure and stochiometry are indispensable tools for subcellular mapping in high-resolution TEM. Electron microscopic imaging exploits the high electron density of gold (19.3 g/ml) compared to that of proteins (1.35 g/ml), providing electron opacity and high contrast to biological materials, and thus guarantees reliable detection during visual or automated evaluation. Due to its particulate and countable nature, colloidal gold is preferred as an antibody label as it offers the ability to quantify the concentration of antigens. However, specificity and affinity of the antibody can influence nanoparticle quantification measurements. Accordingly, the number of nanoparticles cannot be directly derived from the number of antigens on the sample section, because the labeling efficiency is influenced by various physical, chemical and biological factors, mainly arising from sample preparation. However, constant and reproducible labeling efficiencies achieved by standardizing processing conditions are usually considered sufficient for quantification purposes. In our study, gold nanoparticles that carried a single binding site for the primary antibody. Therefore, we hypothesize that although relative immunogold labeling does not provide the exact antigen concentration, it allows direct comparisons between subcellular site concentrations.”

Arbitrary thresholding is prone to bias. The shape and the area can strongly vary depending on the settings used. This part has to be explained in more detail.

Line 396-413: “In traditional post-embedding TEM experiments, cells are chemically fixed, dehydrated and embedded in resins. Resin blocks containing the specimen are then sectioned into thin slices to ensure collection of the electron beam after passing through the sample. Since biological specimens (cells and tissue) are composed of elements with low atomic numbers, the difference in electron density is small, resulting in low-contrast images. To increase specimen´s contrast, biological samples are traditionally stained with heavy metal salts, such as osmium tetroxide, lead citrate and uranyl acetate. Osmium interacts with lipids, uranium binds to phosphate and amino groups, and lead interacts with negatively charged groups. Overall, these metallic dyes enable differential staining of organelles and compartments in mammalian cells. To ensure meaningful comparisons of the complex organization within cells at the ultrastructural level, all samples must be processed exactly in the same way. Moreover, threshold values ​​for determining LEDDs must not be changed in original TEM images, otherwise the shape and area of LEDDs would have been variable depending on the setting. Another important point is that uranyl acetate contrasting is not specific to DNA and therefore is not a reliable marker for DNA compactness in TEM imaging applications. In future projects systematic investigations are planned to analyze the pathophysiological significance of these LEDDs using immunogold-labeling for euchromatic and heterochromatic histone modifications.”

Reviewer 2 Report

The paper presents an interesting application of automated image analysis to quantify radiation-induced DNA damage and chromatin changes at the nanoscale using transmission electron microscopy (TEM).  Comparative analysis of low-LET vs high-LET irradiated cells provides insights into differences in DNA damage patterns and chromatin remodeling dynamics. Overall, the study demonstrates a valuable methodology to gain new biological insights.

However, this paper should be further improve in following part before accept:

Introduction

  • Add more background on the limitations of conventional techniques to characterize complex DNA damage patterns. This will better motivate the need for the current approach.
  •  

Methods

  • Provide more details on the specific algorithms, software tools and parameters used for image processing, segmentation and analysis.
  • Include images showing intermediate steps in the automated analysis workflow. This will help readers better understand the process.
  • Discuss any limitations of the automated analysis such as need for manual corrections. How was accuracy ensured?

Results

  • For nanoparticle cluster analysis in Fig 2B and 3B, add statistical tests to quantify significant differences between groups.
  • Perform spatial distribution analysis around DCRs for other repair factors besides 53BP1 to strengthen conclusions.
  • Validate automated quantification against manual counting for a few sample images to demonstrate efficiency gain.
  • Use clear and consistent formatting for all figures - fonts, sizes, axis labels etc. 

Discussion

  • Compare the information gained from the automated TEM analysis versus conventional IFM imaging. What new insights were revealed?
  • Discuss potential future applications/extensions of the automated analysis approach.

Overall

  • Carefully proofread to fix minor language and grammar errors. For example, there is no 0.1 h in Figure 1A
  •  

  • Carefully proofread to fix minor language and grammar errors. 

Author Response

Reviewer #2

The paper presents an interesting application of automated image analysis to quantify radiation-induced DNA damage and chromatin changes at the nanoscale using transmission electron microscopy (TEM).  Comparative analysis of low-LET vs high-LET irradiated cells provides insights into differences in DNA damage patterns and chromatin remodeling dynamics. Overall, the study demonstrates a valuable methodology to gain new biological insights.

However, this paper should be further improve in following part before accept:

Introduction

  • Add more background on the limitations of conventional techniques to characterize complex DNA damage patterns. This will better motivate the need for the current approach. 

Line 371-385: “Radiation-induced DSBs have major effects on cell biology of transcription, replication and interface with metabolic responses. Accurate recognition and timely repair of DSBs in complex chromatin environments requires a tightly coordinated DNA damage response (DDR). To detect DSBs in cell nuclei, IFM is generally used for visualization of γH2AX or other radiation-induced foci. However, the resolution of standard fluorescence microscopy is too low to detect individual proteins at single-molecule level, so DNA repair events cannot be linked to other DDR mechanisms. Our previous TEM studies with nanoscale-resolution imaging of accumulated DNA damage after high-LET IR revealed intriguing new insights into DSB processing within the chromatin environment. The basic idea of ​​this TEM study was not only to go beyond the resolution of IFM, but to systematically record and evaluate the various repair factors related to the chromatin status with a feasible time commitment. Automated image analysis of TEM micrographs offers an unbiased approach to investigate DDR by measuring protein localizations, interactions and concentrations in the ultrastructure of the cell nucleus.“

Methods

  • Provide more details on the specific algorithms, software tools and parameters used for image processing, segmentation and analysis.

Line 387-394: “In this study, we used the computational pathology software HALO AI to evaluate TEM micrographs. HALO AI image analysis platform is specialized for quantitative analysis in digital pathology, enabling segmentation using artificial intelligence. User-friendly, intuitive HALO modules for different applications improve image processing speed and permit transparent workflows. Here, we used automatic segmentation of differently sized nanoparticles in combination with spatial analysis to investigate precisely the immunogold-labeling patterns in nuclear sections, thereby characterizing the DNA damage pattern after exposure to different radiation qualities.”  

However, since HALO AI is a commercially available software, we unfortunately cannot provide any more precise information about specific algorithms used.

  • Include images showing intermediate steps in the automated analysis workflow. This will help readers better understand the process.

There are no further image processing steps in the automated workflow that can be shown.

  • Discuss any limitations of the automated analysis such as need for manual corrections. How was accuracy ensured?

Reliable image analysis results require consistent specimen preparation, stain optimization and image acquisition. Only if this sample preparation is carried out in absolutely the same way, trustworthy results can be achieved with automated image analysis of TEM micrographs.

Line 396-413: “In traditional post-embedding TEM experiments, cells are chemically fixed, dehydrated and embedded in resins. Resin blocks containing the specimen are then sectioned into thin slices to ensure collection of the electron beam after passing through the sample. Since biological specimens (cells and tissue) are composed of elements with low atomic numbers, the difference in electron density is small, resulting in low-contrast images. To increase specimen´s contrast, biological samples are traditionally stained with heavy metal salts, such as osmium tetroxide, lead citrate and uranyl acetate. Osmium interacts with lipids, uranium binds to phosphate and amino groups, and lead interacts with negatively charged groups. Overall, these metallic dyes enable differential staining of organelles and compartments in mammalian cells. To ensure meaningful comparisons of the complex organization within cells at the ultrastructural level, all samples must be processed exactly in the same way. Moreover, threshold values ​​for determining LEDDs must not be changed in original TEM images, otherwise the shape and area of LEDDs would have been variable depending on the setting. Another important point is that uranyl acetate contrasting is not specific to DNA and therefore is not a reliable marker for DNA compactness in TEM imaging applications. In future projects systematic investigations are planned to analyze the pathophysiological significance of these LEDDs using immunogold-labeling for euchromatic and heterochromatic histone modifications.”

Results

  • For nanoparticle cluster analysis in Fig 2B and 3B, add statistical tests to quantify significant differences between groups.

For low-LET and high-LET IR statistical analysis was performed for pKu80 and DNA-PKcs nanoparticles in total (as well as in euchromatin and heterochromatin, respectively) and for differently sized clusters, as described in the Methods section. Differences between low-LET versus high-LET IR are marked by brackets with asterisks.

Line 156-164: “Statistical analysis: GraphPad Prism (version 9.4.1, GraphPad Software, San Diego, CA, USA) was used to analyse data. Data were presented as the mean of at least three experiments ±SE. Two-way ANOVA (multiple comparisons) and Mann-Whitney-Test were used for estimating the differences among groups, followed by multiple comparisons between data sets. A p-value of <0.05 was considered statistically significant, <0.01 as highly statistically significant and <0.001 as exceptionally statistically significant. In the figures, statistically significant differences are indicated as asterisks directly above the bar when comparing to the previous time point and as asterisks abovebrackets when comparing between two different study groups (* p<0.05, ** p<0.01, and *** p<0.001).”

  • Perform spatial distribution analysis around DCRs for other repair factors besides 53BP1 to strengthen conclusions.

Since reviewer 1 doubts whether the DCRs (now called low electron density domains, LEDDs) are actually decondensed chromatin regions, we would first like to clarify this through additional experiments in future projects before we make further efforts in this direction. Moreover, the focus of this work is the technical establishment of automated image analysis for evaluating TEM micrographs.

Line 409-413: “Another important point is that uranyl acetate contrasting is not specific to DNA and therefore is not a reliable marker for DNA compactness in TEM imaging applications. In future projects systematic investigations are planned to analyze the pathophysiological significance of these LEDDs using immunogold-labeling for euchromatic and heterochromatic histone modifications.”

  • Validate automated quantification against manual counting for a few sample images to demonstrate efficiency gain.

Nanoparticle-antibody conjugates with defined structure and stochiometry are indispensable tools for subcellular mapping in high-resolution TEM. Electron microscopic imaging exploits the high electron density of gold (19.3 g/ml) compared to that of proteins (1.35 g/ml), providing electron opacity and high contrast to biological materials such as cells or tissue. As part of the method establishment, it could be shown that following contrast adjustment that, the 100% of nanoparticles were automatically detected and segmented due to this large difference in density, so that there are no differences between visual and automated nanoparticle detection.

Line 415-432: “Nanoparticle-antibody conjugates with defined structure and stochiometry are indispensable tools for subcellular mapping in high-resolution TEM. Electron microscopic imaging exploits the high electron density of gold (19.3 g/ml) compared to that of proteins (1.35 g/ml), providing electron opacity and high contrast to biological materials, and thus guarantees reliable detection during visual or automated evaluation.”

  • Use clear and consistent formatting for all figures - fonts, sizes, axis labels etc. 

The figures have been revised to ensure the formatting is as clear and consistent as possible.

Discussion

  • Compare the information gained from the automated TEM analysis versus conventional IFM imaging. What new insights were revealed?

The basic idea of ​​this TEM study was not only to go beyond the resolution of fluorescence microscopy, but also to systematically record and evaluate the various repair factors related to the potential chromatin status in a feasible time commitment through automated image analysis.

Line 371-385: “To detect DSBs in cell nuclei, IFM is generally used for visualization of γH2AX or other radiation-induced foci. However, the resolution of standard fluorescence microscopy is too low to detect individual proteins at single-molecule level, so DNA repair events cannot be linked to other DDR mechanisms. Our previous TEM studies with nanoscale-resolution imaging of accumulated DNA damage after high-LET IR revealed intriguing new insights into DSB processing within the chromatin environment. The basic idea of ​​this TEM study was not only to go beyond the resolution of IFM, but to systematically record and evaluate the various repair factors related to the chromatin status with a feasible time commitment. Automated image analysis of TEM micrographs offers an unbiased approach to investigate DDR by measuring protein localizations, interactions and concentrations in the ultrastructure of the cell nucleus. “

  • Discuss potential future applications/extensions of the automated analysis approach.

Line 450-456: “Overall, this automation strategy for quantifying nanoparticles in the chromatin context significantly reduces workload and enables comparative studies to evaluate dose distributions on micro- and nanometer scales following exposure to different radiation qualities. The increased throughput provided by automated acquisition schemes and the resulting generation of large amounts of data opens new possibilities for quantitative TEM studies in radiation research.”

Overall

  • Carefully proofread to fix minor language and grammar errors. For example, there is no 0.1 h in Figure 1A  

This error has been corrected accordingly.

Comments on the Quality of English Language

  • Carefully proofread to fix minor language and grammar errors. 

Anna Isermann, a native English speaker, carefully proofread the manuscript.

Round 2

Reviewer 1 Report

The authors made reasonable corrections. I support acceptance of the manuscript.